# Cell Rearrangement and Oxidant/Antioxidant Imbalance in Huntington’s Disease

**DOI:** 10.3390/antiox12030571

**Published:** 2023-02-24

**Authors:** Francesco D’Egidio, Vanessa Castelli, Annamaria Cimini, Michele d’Angelo

**Affiliations:** Department of Life, Health and Environmental Sciences, University of L’Aquila, 67100 L’Aquila, Italy

**Keywords:** neurodegeneration, oxidative stress, diet, vitamins, mitochondria, huntingtin, impairment, neuron, ROS

## Abstract

Huntington’s Disease (HD) is a hereditary neurodegenerative disorder caused by the expansion of a CAG triplet repeat in the *HTT* gene, resulting in the production of an aberrant huntingtin (Htt) protein. The mutant protein accumulation is responsible for neuronal dysfunction and cell death. This is due to the involvement of oxidative damage, excitotoxicity, inflammation, and mitochondrial impairment. Neurons naturally adapt to bioenergetic alteration and oxidative stress in physiological conditions. However, this dynamic system is compromised when a neurodegenerative disorder occurs, resulting in changes in metabolism, alteration in calcium signaling, and impaired substrates transport. Thus, the aim of this review is to provide an overview of the cell’s answer to the stress induced by HD, focusing on the role of oxidative stress and its balance with the antioxidant system.

## 1. Introduction

Huntington’s Disease (HD) is a neurodegenerative disease caused by a cytosine–adenine–guanine (CAG) triplet expansion mutation in exon 1 of huntingtin gene (*HTT*) responsible for the degeneration of striatal neurons in the brain. The *HTT* gene is sited in the short arm of chromosome 4 (4p.16.3). The result of the mutation is the abnormal expansion of the polyglutamine (polyQ) tract in the Htt protein. From a clinical point of view, HD shows several symptoms, among which are cognitive disorders, chorea, and progressive dementia, leading to death in approximately 15–20 years [1]. A consequence of the genetic alteration is the production of a mutant huntingtin protein (mHtt). The toxic gain of function of mHtt has been considered the main feature of the pathology, with loss of advantageous tasks of the normal huntingtin protein (Htt) followed by numerous alterations at cellular level. 

HD is a fully penetrant neurodegenerative disease caused by a dominantly inherited CAG trinucleotide repeat expansion. Repeats of 36 or more triplets are pathogenic, with longer repeats typically causing earlier onset. When the number of CAG triplets is between 36 and 39, penetrance is reduced, probably due to the disease onset being beyond the normal lifespan in people carrying these alleles [2]. When the gene is inherited by the paternal line, the anticipation phenomenon can also be observed, resulting in a pathogenic expanded polyQ tract in a child whose father had a CAG repeat length in the intermediate range. In fact, the CAG repeat is more instable in male sperm than in somatic tissues. This expansion can be observed in people at risk through a predictive genetic test [3,4]. However, genetic diagnosis is not the only solution. 

In fact, HD can be diagnosed clinically by observing characteristic signs and symptoms. For instance, in the early phase of the disease, neuronal loss characterizes the cortex and striatum. With the progression of HD, globus pallidus, *substantia nigra*, thalamus, sub-thalamic nucleus, hypothalamus, and cerebellum are affected by neurodegeneration [5,6,7,8,9]. In particular, GABAergic Medium Spiny Neurons (MSNs) in the striatum and cortex manifested higher vulnerability to excitotoxicity caused by mHtt [10]. In the advanced phases of the disease, frontal and temporal lobes show neuropathological features [11]. From a clinical point of view, the manifestation of the neuropathological condition consists of motor, cognitive, and behavioral symptoms. The motor impairment manifests as unintentional and out of control spasmodic movements defined as “chorea”, but also as rigidity and physical instability, difficulty in speaking, mastication, and swallowing [12,13]. Altered motor behavior manifests as a consequence of neurodegeneration of MSNs in the striatum. HD shows a peculiar motor feature. In fact, in the early phases of the disease, patients with HD experience hyperkinesia. On the contrary, in later phases, patients experience hypokinesia. The switch from hyperkinesia to hypokinesia reflects the balance existing between the direct and the indirect pathway related to the dopamine receptor 1 and 2 (D1R and D2R), expressed by MSNs principally in the limbic system, striatum, thalamus, and hypothalamus [14,15,16]. At the early phase of HD, hyperkinesia manifests due to the reduction in D2R levels that results in the disruption of the indirect pathway and over-activation of the motor cortex. As the disease progresses to later phases, the lower levels of D1R lead to over-inhibition of the motor cortex with disruption of the direct pathway, via thalamus, resulting in hypokinesia. Thus, the disruption of the balance between the direct and the indirect pathway in MSNs results in the switch between hyperkinesia and hypokinesia, often reaching rigidity in the final phases of the disease [15,16,17,18]. Regarding cognitive symptoms, dysfunction of frontal–subcortical neuronal circuitry and subcortical atrophy are responsible for impaired planning ability, abstract thinking, judgement, flexibility, and rule acquisition [12]. Degeneration of the hippocampal region of the frontal cortex and of the striatum is responsible for not only cognitive symptoms but also motor impairment [19,20,21]. Moreover, degeneration in the cingulate cortex is associated with mood changes [22]. Behavioral symptoms include irritability, depression, addictive and compulsive behavior, anxiety, and anger. Symptoms proceed inexorably once they manifest, until the death of the patient. To date, there is no cure available for HD, but multidisciplinary management of signs and symptoms seems to ameliorate patients’ quality of life. 

### 1.1. HD Epidemiology

Epidemiological data of HD are dramatically influenced by genetic evaluation of the CAG repeat expansion. In fact, HD’s prevalence estimation depends on the coupling of a genetic test and neurological evaluation of the onset of the disease. Such coupled studies showed different prevalence estimates across the globe. HD occurs worldwide, with a prevalence of ~13 individuals per 100,000 in Western populations [23,24,25]. Interestingly, prevalence studies based on genetic tests reported higher rates of HD than those based on clinical diagnosis alone [26]. The increasing prevalence of HD coincides with the wider availability of the molecular test [23,27]. Before the genetic test, family history was the diagnosis criterion. Due to this, exclusion of de novo or sporadic cases occurred. The molecular diagnosis led to information recovery about excluded cases. Moreover, the genetic test proved the presence of late-onset HD in the elderly population, often characterized by lack of family history or challenging clinical diagnosis due to the simultaneous presence of dementia and other neurodegenerative diseases. 

Regarding incidence, HD shows ~5.8 new cases every million per year in Western populations [28,29]. Although HD is present worldwide, the disease occurs with a lower frequency in Japan and China, spanning from one to seven people every million, compared to the higher frequency of Europe and North America. Additionally, in South Africa, HD frequency is lower in black individuals than in white populations. In particular, HD is more frequent among people with European ancestry [26]. Local clinical reviews and case studies are the only source of epidemiological data from other populations in Asia and Africa. In addition, various isolated areas of high prevalence have been documented. Of particular interest is the Maracaibo area in Venezuela, which is characterized by the presence of hundreds of HD patients traced to a single ancestor. Nevertheless, the overall worldwide prevalence and incidence rates of HD remain unclear [3,30].

### 1.2. HD Pathogenesis

Decades have passed since the discovery of the causative mutation of HD in 1993 [31]. Still, the physiological role of Htt and its normal function, as well as the principal pathogenic mechanisms of the expansion mutation, remain undefined. Extensive research produced data suggesting the involvement of Htt in various processes inside cells, such as DNA maintenance, axonal trafficking, antiapoptotic activity, transcription regulation, cell signaling, and energy metabolism modulation. Furthermore, a vital role for Htt has been proposed. In fact, without Htt, life is not possible for mammals. An essential role for Htt has been highlighted in embryonic development [32,33,34,35,36]. However, when HD occurs, mHtt affects all those cellular processes in which its wild-type forms are involved, leading to an overall cellular dysfunction characterized, among all, as described by Figure 1 as the accumulation of toxic proteins, alteration of nuclear pore complex, changes in DNA methylation, mitochondrial metabolism impairment, damaged proteins and organelles clearance system dysfunction, but also transcriptional dysregulation, DNA repair system alteration, excitotoxicity, altered vesicle transport, loss of neurotrophin support and neuroinflammation, and oxidative stress [37].

In this section, the major pathogenic features of HD will be reported based on the most recent studies. Among all these features, cell rearrangements and oxidative stress will be dissected in the following sections.

The mHtt has been considered one of the major actors in HD pathophysiology. The expansion of the CAG triplet in the HD gene is responsible for the production of an aberrant protein characterized by a prolonged polyQ tract. The expanded tract prevents the normal fold of mHtt, resulting in an unfolded soluble protein. The soluble monomers of mHtt combine in oligomers that are involved in the formation of mHtt fibrils and large inclusions in the cytoplasm and nucleus, which were considered to be pathogenic [38,39,40]. However, recent studies showed that mHtt inclusions can occur not only with cell death but also without cell death [38,41,42]. Moreover, a toxic role for N-terminal mHtt oligomers has been proposed [43,44,45,46,47,48]. The origin of these oligomers relies on proteolytic cleavage of mHtt but also on a CAG length-dependent aberrant splicing that produces a short mRNA which is translated into the toxic N-terminal fragment containing exon 1 [2,39,49,50]. The growing availability of toxic fragments can be a dangerous characteristic. Thus, a protective effect derived from the formation of inclusions has been suggested as a scavenging system able to reduce toxic fragments’ availability [41,44]. In this context, endoplasmic reticulum (ER) stress that usually precedes mHtt aggregation shows a critical role in improving the formation of larger inclusions [51,52]. Evidence suggests that toxic N-terminal fragments aggregate more rapidly in the brain of patients with HD than the full-length protein does [53,54,55]. Moreover, the ability of mHtt to transfer between cells has been suggested. In vitro models of HD showed synthetic polyQ peptides’ uptake in cell cultures but also the transfer of fluorescent mHtt between cells [56,57,58,59]. In *Drosophila*, mHtt released from synaptic terminals can be taken up by endocytosis by adjacent neurons [60,61]. The contiguous spreading through functional synapses has been observed in other HD models [62]. However, evidence of mHtt propagation in human is limited. Fibrils and oligomers of mHtt have been found in the brain of patients with HD [63,64]. *Post-mortem* investigation of the graft in patients that received fetal striatal transplants suggested that mHtt is released by neurons due to the presence of inclusions in the extracellular matrix. Still, inclusions were not found within cells [65].

Toxic mHtt was found to be able to disrupt the nuclear pore complex. The complex is an active transport system between the nucleus and cytoplasm through which RNA and proteins are transported [66]. In this system, the ras-related nuclear protein (Ran) is involved. Ran is converted inside the nucleus from Ran-GDP to Ran-GTP by the regulator of chromosome condensation 1 (RCC1) on the cytoplasmic filaments of the nuclear pore complex back to Ran-GDP via interaction with Ran GTPase-activating protein 1 (RanGAP1). In this way, a critical concentration gradient of the Ran forms is established, acting as a signal for cellular processes. In HD, the mHtt showed a greater affinity to RanGAP1 compared to the wild-type protein. Thus, the alteration of the balance between the Ran’s forms caused by mHtt interactions with RanGAP1 can lead to the loss of the nuclear to cytoplasmic Ran’s gradient produced by RanGAP1, quickly resulting in cell death [67,68,69].

DNA methylation, as an epigenetic mechanism, plays a relevant role in the regulation of gene expression both in the physiological condition and in HD. It involves the formation of 5-methylcytosine through the addition of a methyl group on the pyrimidine ring of cytosines catalyzed by a class of enzymes called DNA methyltransferases at promoter, proximal, and distal regulatory regions, occurring both at cytosine–phosphate–guanine (CpG) and non-CpG islands [70,71]. In the HD context, the toxic protein is able to regulate gene expression. In fact, mHtt can interfere with the transcriptional machinery by modulating the post-transcriptional modifications of histones and DNA methylation, resulting in altered gene expression and neuronal dysfunction. In particular, modulation of DNA methylation levels was found at CpG-rich regions located near the transcription start sites, with increased or decreased methylation levels [72,73,74]. In these studies, the resulting methylation was inversely correlated with the expression of genes involved in several processes, among which are neuronal migration, cell differentiation, and signal transduction [75,76]. DNA methylation has been also suggested as a biomarker in HD. For example, a role for mHtt has been proposed in increasing the biological age of cells and tissues in which it is expressed. In this regard, using the method of the “epigenetic clock”, a DNA methylation-based biomarker of tissue age, the correlation between HD and aged brain tissues has been observed. Specifically, HD association with epigenetic age acceleration and aberrant DNA methylation has been demonstrated, reflecting the great effects of chronological age on DNA methylation levels [77,78,79,80,81]. Moreover, the epigenetic clock method allowed the correlation between manifest HD with the increment of the epigenetic age to be observed in human blood DNA, with DNA methylation levels associated with motor score progression in HD patients [82].

Mitochondrial metabolism is essential for neuronal survival. Nevertheless, mitochondria are involved in HD pathogenesis. Several studies showed that mHtt can directly affect mitochondrial functions, resulting in the disruption of calcium handling via direct interaction with the mitochondrial outer and inner membrane and the aberrant transcription of nuclear genes involved in proper mitochondrial functioning [83,84,85,86,87]. The overall impairment of mitochondria leads to dysfunctional respiration. 

In *post-mortem* brain samples of patients with HD, increased mitochondrial DNA mutations, reduced ATP production, and mitochondrial ultrastructure disruption have been found [88,89]. Moreover, other features of *post-mortem* brain samples were a lower number of mitochondria and reduced activity of enzyme complexes [90,91,92,93]. In animal models of HD, the expression of the peroxisome proliferator-activated receptor gamma coactivator 1-alpha (PGC-1α), which is involved in the regulation of mitochondrial biogenesis, was lower than in normal controls [91,94]. In addition, an interplay between PGC-1α and p53, a well-known transcription factor mainly involved in tumor suppression, has been observed, leading to mitochondrial dysfunction-derived oxidative stress but also inflammation, neurodegeneration, and apoptosis [95]. Moreover, mHtt affected mitochondrial axonal anterograde and retrograde transport, resulting in accumulation of mitochondria in the soma [96,97,98,99].

The accumulation of toxic proteins in the cytoplasm is normally counteracted by different cell systems. The ubiquitin–proteasome system is responsible for the clearance of damaged proteins, whereas autophagy degrades damaged organelles and protein complexes. In HD, the impairment of the ubiquitin–proteasome system and autophagy has been suggested [100,101].

## 2. Cells Rearrangement in HD

The accumulation of mHtt inside cells leads to an overall dysfunction of the cellular machinery. Neuronal cells are particularly vulnerable to the various effects of toxic protein accumulation. Normal neurons try to adapt to the various alterations induced by stressful conditions with rearrangement in organelle dynamics and modulation of the metabolism. In HD, the disruption of these systems has been described.

Failure in the activation of ER response or in unfolded protein response (UPR), involved in the refolding of misfolded proteins via molecular chaperones or in the degradation via proteasome system, has been described in HD. The toxic mHtt oligomers are able to interfere with ER-associated degradation (ERAD) components [51,102]. Normally, ER stress is responsible for the activation of UPR. Its activation can occur through three UPR sensors, which are transmembrane proteins called inositol-requiring protein 1 α (IRE1), activating transcription factor 6 (ATF6), and PKR-like ER-localized eIF2α kinase (PERK) [103]. IRE1 is a kinase associated with the Binding immunoglobulin Protein (BiP), a molecular chaperone by which IRE1 dissociates upon growing levels of unfolded proteins. Autophosphorylation of IRE1 led the protein to switch to its active form which is responsible for the splicing of the mRNA encoding X-box binding protein 1 (XBP-1). XBP-1 is a transcription factor that, once translated, migrates to the nucleus to induce the expression of BiP and other ERAD components [104]. 

Regarding ATF6, it is a transcription factor associated with BiP which translocates from the ER to the Golgi upon BiP dissociation. In the Golgi, ATF6 is detached from the Golgi membrane after protease-mediated cleavage. In this way, the cytoplasmic domain of ATF6 can move to the nucleus to stimulate the expression of UPR genes, among which are BiP and XBP1 [105,106,107]. PERK is associated with BiP, as well. PERK dissociates from BiP and switches to its active form by autophosphorylation. In this way, the sensor protein can phosphorylate the nuclear erythroid 2 p45-related factor 2 (NRF2). Active NRF2 dissociates from the complex formed with Keap1 and translocates to the nucleus in order to stimulate the expression of genes related to the antioxidant response [108]. When ER stress occurs, the three UPR sensors lead to an increase in chaperones production and enhanced degradation, as well as protein translation inhibition via eIF2α. However, if protein homeostasis cannot be restored, the chronic activation of UPR starts an apoptotic pathway that results in cell death [109,110,111,112]. In HD, the inhibition of ERAD and the impairment of the stress responses have been observed in several in vitro and in vivo models, but also in post-mortem samples [51,102,113,114,115,116,117,118]. Evidence has suggested that the interference caused by mHtt relies on the depletion of important proteins related to the ubiquitin-proteasome system, such as p97/VCP and the cofactors UFD1 and NPL4, but also USP14 and ATF5, leading to ERAD impairment and ER stress response failure [51,102,119,120,121].

In HD, defective mitochondria and altered metabolism are main features of the pathogenesis. Aggregates of mHtt spread within the cells, also reaching mitochondrial membranes where they cause oxidative damage. In fact, mHtt is able to directly interact with the mitochondrial outer membrane, triggering the release of Ca^2+^ and causing alteration in mitochondrial trafficking and morphology. The general dysfunction of mitochondria caused by mHtt’s interaction with the organelle’s outer membrane leads to the disruption of the mitochondrial permeability transition pore, with swelling and depolarization of mitochondria, often ending with cell death [122,123]. Evidence has suggested that mitochondria are more susceptible to the stress induced by Reactive Oxygen Species (ROS) or Ca^2+^ in neurons expressing mHtt due to the incapacity to counteract their accumulation [124,125,126]. Several cellular studies found lower levels of complex II activity in the respiratory chain in samples from affected brain regions of a patient with HD [124]. Moreover, the expression of mHtt leads to higher levels of lactate in cortex and basal ganglia and reduced mitochondrial membrane potential in patient lymphoblasts [84]. Mitochondrial size and numbers were also found to be reduced in post-mortem HD brain samples, with changes in mitochondrial ultrastructure such as deranged crests, altered matrix, and vacuolization [127,128]. 

Mitochondria promote cell death when the damage received becomes irreversible. Normally, mitochondria undergo the fission process, regulating the release of proteins of the intermembrane space, such as cytochrome c, to facilitate apoptosis [129]. The signaling pathways that regulate the fission comprise proteins such as Huntingtin-Interacting Proteins 1 and 14 (HIP1 and HIP14), clathrin, endophilin 3, and dynamin. However, in HD, the mHtt directly binds to dynamin-related protein 1 (Drp1), resulting in the activation of the fission process [130,131,132]. The fission process has been suggested to be the predominant solution for neurons in HD pathology. In fact, higher levels of Drp1 and mitochondrial fission 1 (Fis1) proteins were found along with reduced levels of mitofusins 1 and 2 (Mfn1/2), resulting in extensive fragmentation [90,98,133]. In this context, accumulation of damaged mitochondria can occur, leading to activation of a selective mitochondrial degradation system called mitophagy. The mitophagy can be dependent or independent of the phosphatase and tensin homolog (PTEN)-induced putative kinase 1 (PINK1)/Parkin pathway [134,135]. However, a role for Htt has been proposed in the regulation of autophagosome dynamics through the control of kinesin and dynein along with huntingtin-associated protein 1 (HAP1). In HD, axonal transport of autophagosomes resulted in inefficient degradation of internalized mitochondria in the presence of mHtt. The normal protein can act as a scaffold protein in mitophagy. Probably, the mHtt cannot carry out this task, resulting in the inhibition of autophagosome/lysosome fusion [136,137,138].

## 3. Oxidative Stress 

Stress conditions induced by HD pathology find a common thread in oxidative stress. The cell rearrangement in cells affected by HD can be preceded or followed by oxidative stress. For instance, the mitochondrial function is affected by ER stress, leading to the exacerbation of an already present oxidative stress [139]. 

The growing levels of oxidants compared to those of antioxidants lead to an imbalance condition called oxidative stress. The increase in oxidants causes cell dysfunction and damage, often ending with cell death. The impaired redox homeostasis is characterized by growing levels of reactive species, short-lived molecules, among which are free radicals, responsible for oxidative damage to any cellular element, in particular DNA, proteins, and lipids. Oxidants are delineated by the presence of unpaired electrons in the outer orbitals, resulting in extremely reactive molecules able to take electrons from neighboring molecules. However, oxidants are produced by cells in different forms, such as ROS, reactive nitrogen species (RNS), or reactive lipid species (RLS) [140]. The principal source of ROS is mitochondria in mammalian cells due to their high levels of oxygen consumption. Through the electron transport chain, mitochondria generate a proton gradient and produce ATP. During the process, a tiny part of the electrons destined to energy production interacts with oxygen molecules, producing anion superoxide free radicals (O2-) that quickly interact with neighboring biomolecules and produce more reacting molecules, among which are peroxyl radical (ROO·), hydroperoxyl radical (HOO·), hydrogen peroxide (H_2_O_2_), and peroxinitrite radical (ONOO·) [141,142]. In particular, anion superoxide accumulation leads to Fe^2+^ release from Fe-carrying molecules. In this way, iron can freely undergo the Fenton reaction with H_2_O_2_ producing, among all, hydroxyl radical (_OH), a highly reactive compound. Moreover, the interaction between anion superoxide and peroxinitrite is responsible for the production of nitric oxide (NO) [142,143,144]. Thus, reactive species, such as ROS and RNS, are produced by mitochondria during normal metabolism as byproducts [140,145]. However, mitochondria are not their only intracellular producers. For instance, ROS can be produced by NADPH oxidases of the plasma membrane but also by xanthine oxidase, peroxisomal flavin oxidases, and cytochrome P450 enzymes of the ER [146,147,148]. Regarding RLS, they are electrophilic compounds derived from polyunsaturated fatty acids (PUFAs) oxidation, such as malondialdehyde, acrolein, hydroxynonenal (HNE), isoprostanes, etc. PUFAs can be oxidized by enzymes such as cyclooxygenase or lipoxygenase, but there are also non-enzymatic ways responsible for RLS production [149]. RLS are able to covalently alter several molecules, among which are proteins, lipids, DNA, and RNA [149,150]. 

When the imbalance between oxidants and antioxidants delineates and oxidative stress occurs, the cell machinery tries to restore physiological homeostasis. However, ROS, RNS, and RLS have a detrimental effect when their levels grow higher, resulting in increased metabolic rate, higher levels of transition metals, reduced antioxidant levels, etc. For instance, an excess of reactive molecules damages mitochondria. ROS and RNS produce lipid peroxidation interacting with mitochondrial membranes, resulting in RLS production. In this way, RLS amplifies oxidative stress, stimulating ROS and RNS production [128,149,150,151].

### Oxidative Stress in HD

Oxidative stress plays a major role in HD pathology. In particular, neurons are extremely susceptible to oxidative damage. MSNs from striatum exhibited, among all, greater vulnerability to toxic protein accumulation and following effects. The cause of MSNs’ sensitivity relies on the extra-synaptic glutamatergic signaling [152]. Normally, the release of glutamate from neuronal synapses activates the N-methyl—D-aspartate receptors (NMDARs) that induce Ca^2+^ uptake promoting neuronal plasticity and survival [153]. In HD pathology, mHtt is responsible for the imbalance between synaptic and extra-synaptic NMDARs signaling, impairing the neuroprotective cascade related to cyclic AMP response element-binding protein (CREB) and PGC-1α. In this context, the deregulation of PGC-1α expression leads to reduced expression of antioxidants, such as superoxide dismutase 1 and 2 (SOD1 and SOD2) as well as glutathione peroxidase 1 (GPx1), promoting oxidative stress and cell death [139,152,154,155,156]. However, NMDARs are also involved in other mechanisms related to oxidative stress in mHtt-expressing neurons. For instance, NMDARs mediate Ca^2+^ mitochondrial uptake caused by mHtt interactions with mitochondria, stimulating production of ROS and damaging the mitochondrial DNA [157]. The impaired calcium signaling derived from mHtt interactions with mitochondria has detrimental effects on organelles’ dynamics [158,159,160]. Damaged mitochondria play a crucial role in amplifying the oxidative stress, but ER, the ubiquitin–proteasome system, and autophagy are also involved. As already mentioned above, mHtt can alter these systems, causing oxidative stress, higher levels of metal ions, and impairing protein homeostasis [88,161,162,163,164,165,166]. 

However, oxidative stress does not relate only to inner cell mechanisms. A critical role has been suggested for oxidative stress in the progression of HD’s neuroinflammation. In this regard, major actors are the glial cells, among all, astrocytes and microglia. As principal modulators of inflammation, astrocytes and microglia are non-neuronal support cells in the nervous system [167,168]. Astrocytes are the most abundant glial cell in the human brain [169]. In physiological conditions, astrocytes can be found as resting astrocytes or reactive astrocytes [170]. Resting astrocytes are involved in various processes, among which are the regulation of intercellular space conditions, synaptic formation and maintenance, and neurovascular activities as part of the Brain–Blood Barrier (BBB) [170,171]. When a proinflammatory stimulus occurs, astrocytes undergo reactive astrogliosis. In this process, they can become protective astrocytes or harmful astrocytes [172,173]. As reactive astrocytes, they are responsible for the production of both anti- and pro-inflammatory cytokines, but also of higher intercellular levels of ROS and potassium [172,174]. Regarding microglia, these glial cells play an important immune role but are able to accomplish many other tasks, from the creation of neural networks through brain-derived neurotrophic factor (BDNF) release to synaptic maintenance [168]. Microglia can also be found in two states: the surveilling state, in which microglia work as a chaperone for neighboring neurons, or the activated state. The switch between the two states can take place when an inflammatory stimulus occurs, including neuronal death induced by mHtt. Activated microglia is a major actor of neuroinflammation. In the activated state, microglia can digest exogenous bodies and produce ROS and pro-inflammatory cytokines, but also quinolinic acid, inducing toxicity to selected targets [175]. In several in vivo models of HD and in affected patients, activated glial cells have been observed [170,172,175,176,177]. In particular, mHTT showed itself to be able to modulate the NF-kB pathway, stimulating the release of interleukins (ILs) such as IL-4, IL-6, IL-8, IL-10, and Tumor Necrosis Factor α (TNFα) [178]. When activated, these cells also release oxidizing molecules able to damage the neighboring neurons. In HD, neurons expressing mHtt that are already damaged by internal oxidative stress become targets for intercellular inflammatory and oxidizing agents. In this way, signals from intracellular and intercellular oxidative stress and inflammation activate more glial cells, leading to abnormal glial response. In this cycle of events, oxidative stress and neuroinflammation become chronic over time [174,179].

## 4. Antioxidant Defense and Antioxidant Compounds in HD

As the imbalance in favor of oxidants responsible for oxidative stress continues, the cell machinery tries to counterbalance it with the antioxidant defense. The production of a highly reactive molecule can be counteracted by the scavenging of these molecules. For this purpose, cells play out enzymatic and non-enzymatic strategies [180,181,182]. The enzymatic method involves enzymes such as SOD, GPx, glutathione reductase (GR), and catalase (CAT). The first defense line is SOD. The dismutation reaction catalyzed by SOD allows superoxide anion radical to be reduced into H_2_O_2_. Moreover, H_2_O_2_ can be transformed by CAT and GPx into water and molecular oxygen, resulting in H_2_O_2_ detoxification. Another relevant role is played out by GPx. In fact, GPx can reduce lipid and non-lipid hydroperoxides, peroxidation intermediates, consuming one reduced glutathione (GSH) that will be oxidized into glutathione disulfide (GSSG). GSSG can be then reduced by GR with NADPH as cofactor [183,184,185]. The non-enzymatic strategy may involve several compounds that can be produced by cells during metabolism or need to be introduced by diet. In this regard, glutathione, coenzyme Q10 (CoQ10), bilirubin, L-arginine, uric acid, melatonin, transferrin, and other metal-chelating proteins can be defined as metabolic antioxidants. On the other hand, the main antioxidants provided by diet are vitamins C and E, manganese, zinc, selenium, omega-3 and omega-6 fatty acids, carotenoids, and flavonoids [186]. 

Knowing the role of oxidative stress in HD pathogenesis, several antioxidant compounds have been proposed as potential treatments for this disease. To exert its therapeutic effects in neurodegeneration, an antioxidant must first cross the BBB. However, hydrophilic molecules can cross the BBB only through dedicated carriers. In fact, only tiny levels of hydrophilic molecules are able to pass the barrier, due to BBB’s tight junctions. Conversely, lipid layers of BBB let lipophilic compounds through the barrier when they are of small molecular weight [187]. For these reasons, many antioxidants are unable to cross the barrier. Their charge, the large size, or their polarity negatively influence the neuroprotective effects of antioxidants. Antioxidants can be administered as single molecule or in combination, often as an adjuvant therapy in HD treatment. In several studies, the effectiveness of antioxidants in HD animal models has been observed in regard to disease progression and neurodegeneration. However, clinical trials showed contrasting data with partial neuroprotective effects. In the following sections, various tested antioxidants will be discussed, focusing on their neuroprotective effects (Table 1). 

### 4.1. Vitamin C 

Vitamin C, or ascorbic acid, is an antioxidant synthetized through the pathway of glucuronic acids consuming glucose in animals. However, during evolution, this pathway has been lost by man. Given the impossibility for man to produce Vitamin C, dietary intakes of this vitamin are necessary. In fact, its deficiency causes bleeding from the wrist, ankles, and knees and myalgia and arthralgia as scurvy disease expression. Vitamin C is involved in various functions, such as collagen synthesis, modulation of neuronal metabolism and neuroprotection, and immunological function, but also maintenance of vascular cells’ integrity [238,239,240]. Ascorbic acid is released in the extracellular space during synaptic activity and then neurons again take it up through the Vitamin C transporter, a sodium-dependent specific transport for Vitamin C [241,242]. In the presence of ROS, Vitamin C quickly oxidizes to dihydroxy ascorbic acid, acting as a regulator of redox balance in neurons [243]. In HD, administration of Vitamin C to rodent cortical neurons treated with glutamate counteracted the glutamate-induced neurodegeneration [244]. Moreover, in a R6/2 mouse model, regular administration of ascorbic acid during the onset of behavioral alterations restored the behavioral response [188]. Thus, these data suggest that treatment with Vitamin C may improve the oxidative stress condition that characterizes HD in both in vitro and in vivo models of HD. However, more studies are needed to further understand the role of Vitamin C in HD pathogenesis.

### 4.2. Selenium 

Selenium is a trace essential element involved in the antioxidant activity of selenoproteins. In particular, selenium modulates intracellular redox homeostasis, acting mainly as a cofactor for GPx. In HD rats, selenium showed several effects, such as GABA depletion, improvement in the circling behavior induced by the treatment with quinolinic acid, a NMDA antagonist, and improvement in neuronal morphology. Moreover, selenium pretreatment highly induced GPx activity and reduced lipid peroxidation in rats’ striatum [191]. Selenium showed neuroprotective activity with involvement in neurotransmission. Selenium, but also selenoproteins, counteract inflammation in a neurotransmission context, influencing ion channel activity, brain cholesterol metabolism, calcium homeostasis, and phosphorylation of proteins [245]. In a N171-82Q HD mouse model, sodium selenite supplementation decreased brain weight loss and decreased GSSG levels, but also ameliorated motor function [193].

### 4.3. Unsaturated Fatty Acids 

As an essential component of the diet, unsaturated fatty acids are commonly found in nature. Several studies demonstrated the antioxidant and anti-inflammatory properties of these compounds. For instance, a lower age-dependent neurodegeneration in individuals that maintain a Mediterranean diet has been suggested by many studies, due to the choice of olive oil as a principal fat source [246]. In fact, extra-virgin olive oil (EVOO) exerts various beneficial effects thanks to its high content of oleic acid and other monounsaturated acids, but also phenolic compounds and microconstituents such as phenolic compounds [247,248,249,250]. The major role of EVOO is played in the brain against oxidative stress [251]. A decreased lipid peroxidation and GSH recovery have been found in several brain areas after EVOO administration to 3-nitropropionic acid (3-NP)-induced HD Wistar rats [195]. Moreover, in rats treated with quinolinic acid to induce oxidative stress, a diet rich in fatty acids from EVOO or fish oil showed promising effects counteracting oxidative damage, increasing PPARγ expression, and restoring rat circling behavior [196]. Hence, the neuroprotective effects exerted by EVOO have been widely demonstrated, suggesting the antioxidant role for these compounds in the brain [252,253,254]. Unsaturated fatty acids have also been tested in clinical trials both alone and in combination with other compounds, where a restoration of the motor function has been observed [197,198]. However, more preclinical and clinical trials are needed to assess whether unsaturated fatty acids may play a role in HD management.

### 4.4. Creatine 

Creatine is a natural antioxidant able to neutralize ROS and RNS that exist in cells as free creatine and phosphocreatine [255]. Together, these two forms represent the total creatin amount. In particular, phosphocreatine is required by tissues with high energy demand, such as brain and skeletal muscle, as an energy buffer, where starting from phosphocreatine ADP is phosphorylated into ATP. Creatine kinase is the enzyme responsible for the conversion of creatine into phosphocreatine, which takes place in the presence of ATP that is dephosphorylated into ADP [256]. The creatine role in HD has been the object of discussion for at least two decades. Creatine activity has been tested in several HD animal models but also in clinical trials. For instance, in two different mouse models of HD, R6/2 and N171-82Q, supplementation of creatine extended the animal’s lifespan, improved the motor function, and reduced neuronal atrophy [199,200]. In a rat model of HD (induced with 3-NP), creatine counteracted motor impairment and cognitive abnormalities, suggesting neuroprotective effects of creatine relying on neuronal energy management [201]. 

Regarding clinical trials, two different pilot studies where creatine was administered 10 g/day for 1 year showed no changes in motor and neuropsychological scores [257,258]. Interestingly, in a randomized double-blind placebo-controlled study, creatine was administered 8 g/day for 16 weeks. The group treated with creatine showed, among other features, a reduction induced by creatine of a marker of oxidative damage to DNA called 8-OHdG, normally at high serum levels in HD patients [202]. However, other dosages were also tested. In an open-label add-on study, the observed retarded cortical atrophy suggested the beneficial role of up to 30 g/day of creatine [203]. Nevertheless, in a randomized multicenter double-blind placebo-controlled trial with 553 enrolled HD patients, creatine was administered 40 g/day for 4 years. In this study, creatine was unable to delay the functional decline in HD [204]. Taken together, these data show a solid preclinical base but also mixed clinical results, suggesting the need for a better comprehension of this compound.

### 4.5. Coenzyme Q10 

Coenzyme Q10 (CoQ10), also known as ubiquinone, is an essential player in cell metabolism. It can be found in the inner membrane of mitochondria where it is involved in the activities of complex I and complex II and in ATP production. Several preclinical and clinical studies were conducted with CoQ10 at different dosage regimens. However, no significant changes in the HD phenotype have been observed in subjects treated with CoQ10 alone or in combination with Remacemide [205,207,254,259,260,261,262,263]. In spite of this, coadministration of CoQ10 and creatine in a 3-NP HD animal model showed neuroprotective effects, with restoration of the redox balance and decreased lipid peroxidation and DNA oxidative injury in the cerebral cortex [206]. Moreover, in a R6/2 HD mouse model, coadministration of CoQ10 and creatine promoted survival of mice, improved the motor function, and counteracted the striatal atrophy [203]. The results showed above suggest that coadministration of CoQ10 and creatine results in higher therapeutic potential in HD than using CoQ10 alone.

### 4.6. Idebenone 

Idebenone is a synthetic analog of coenzyme Q10 capable of quickly reaching the brain to exert strong antioxidant activities. Several in vitro and in vivo studies have been carried out with idebenone. Idebenone used as a treatment in rats with neurodegeneration induced by kainic acid intrastriatal injection led to marked restoration of the immunoreactivity of glutamic acid decarboxylase (GAD), a presynaptic striatal marker found to be reduced in this model [208]. On the contrary, the reduction in GAD immunoreactivity has not been counteracted by idebenone in quinolinic acid-induced striatal lesions [264]. However, due to the results obtained, a clinical trial of idebenone has been conducted, enrolling 92 patients with HD. Unfortunately, no meaningful result came out from the trial, showing no impact of idebenone on HD patient conditions compared to the placebo controls [209]. Nevertheless, idebenone has been taken into account in a clinical trial that aims to optimize the symptomatic treatment regimen of Chinese HD patients to improve the prognosis [265].

### 4.7. Curcumin

Curcumin is a lipophilic polyphenol that can be found in turmeric, the curry spice of the ginger family. Curcumin is involved in several functions, among which are antioxidant, neuroprotective, and anti-inflammatory activities. To do so, curcumin regulates various pathways related to cell survival, caspases, tumor suppression, and others [266]. Curcumin has been widely studied on several animal models of neurodegeneration, demonstrating that it is an effective and well-tolerated compound. For instance, curcumin administration suppressed cell death in a *Drosophila* model of HD [211]. In another *Drosophila* model of HD, metabolic anomalies amelioration, ROS levels reduction, and motor impairment counteraction have been obtained after curcumin administration [212]. Curcumin effects have also been tested on a R6/2 HD mouse model where an overall amelioration of the HD phenotype has been obtained [213]. The only problem related to curcumin relies on its bioavailability. In fact, when orally administered, only 25% is available since 75% is lost in feces. Combined treatment with piperine, an alkaloid isolated from the plant *Piper nigrum*, has been suggested to extend the bioavailability of curcumin. Indeed, in a rodent model of HD obtained with quinolinic acid, administration of curcumin in combination with piperine restored the neurochemical alterations and the motor function, also reducing oxidative stress [210].

### 4.8. Grape Seed Polyphenolic Extract 

Grape Seed Polyphenolic Extract (GSPE) is characterized by high concentrations of linoleic acid, flavonoids, Vitamin E, resveratrol, and others. GSPE is a common supplement in the diet that plays a role in limiting lipid peroxidation and counteracting inflammation [267]. In a cell model of HD (PC-12 cells with an inducible protein that contains the first 17 amino acids of Htt plus 103 glutamines tagged with a green fluorescent protein), GSPE reduced the levels of carbonyl heightened by mHtt expression and inhibited the formation of an aggregate of mHtt [214]. The beneficial results obtained in vitro were then confirmed in vivo. In a R6/2 HD mouse model, the reduction in motor impairment and the prolongment of mice lifespan were achieved after treatment with GSPE [214].

### 4.9. Lycopene 

Contained in vegetables and fruits, Lycopene is a phytochemical compound able to quench singlet electrons to exert antioxidant activities. Lycopene has been extensively studied given its potent antioxidant effects. For instance, in HD, lycopene has been tested on 3-NP treated rats. Here, pretreatment with lycopene improved behavioral symptoms and significantly counteracted oxidative damage. In particular, lycopene administration led to restoration of the activity of mitochondrial enzymes, but also reduction in lipid peroxidation, NO and SOD levels, and behavioral impairment [216]. Other studies evaluated in the 3-NP rodent model include the administration of lycopene in combination with other compounds, suggesting that lycopene modulates NO to play an antioxidant role, resulting in restoration of behavioral and biochemical dysfunctions [215,217].

### 4.10. Melatonin 

Melatonin is a hormone produced by the pineal gland characterized by mitochondrial protective effects and scavenging ability of free radicals protecting macromolecules from oxidative damage [268]. In a rat model of neurodegeneration induced with kainic acid, melatonin counteracted the oxidative damage, promoting neuroprotection [220]. In another study, in neurons obtained from a rat model with HD induced by 3-NP, melatonin exerted protective effects counteracting oxidative stress through a reduction in lipid peroxidation and SOD activity [218]. Moreover, in a rat model of HD induced by quinolinic acid, melatonin administration led to behavioral recovery and reduction in ROS, RNS, RLS, and SOD levels [219]. Taken together, these data suggest the neuroprotective role of melatonin in several models of neurodegeneration and, in particular, of HD. In addition, an ongoing clinical trial will evaluate the efficacy in improving sleep quality in HD patients [269].

### 4.11. N-Acetylcysteine 

N-acetylcysteine (NAC) is a naturally found diet supplement with antioxidant properties. NAC, as a precursor of GSH, exerts various activities such as anti-inflammatory and antioxidant activities, regulation of mitochondrial dysfunction and apoptosis, mucolytic activity, stabilization of protein structures, and others [270,271]. In a rodent model of HD, NAC pretreatment showed protective effects counteracting oxidative damage induced by 3-NP [221,272]. In addition, NAC has been suggested to reverse mitochondrial impairment and neurobehavioral effects induced by 3-NP in HD rats [222]. In a R6/1 HD mouse model, NAC exerted its biological activity by counteracting depression-like behaviors in a forced-swim test, showing once more the neuroprotective potential of NAC for HD management [223]. From a clinical point of view, a phase two randomized placebo-controlled study on patients with premanifest HD will evaluate efficacy of the oral administration of NAC [273].

### 4.12. Rutin 

Rutin, a plant pigment found in certain vegetables and fruits, is the glycoside of quercetin exerting antioxidant, anti-inflammatory, cryoprotective, neuroprotective, and immunomodulatory activities [274,275,276,277,278,279,280,281]. In a model of neurodegeneration induced with 3-NP on Wistar rats, rutin exerted protective effects counteracting behavioral anomalies, mitochondrial dysfunction, and oxidative stress [225]. In a following study with the same model, pretreatment with rutin restored the motor function and heightened the levels of non-enzymatic antioxidants, demonstrating once more how it could be a reliable compound for HD management. However, due to the lack of preclinical and clinical studies, further studies are necessary to better define whether and how rutin may be effective against HD.

### 4.13. Tauroursodeoxycholic Acid 

Tauroursodeoxycholic acid (TUDCA) is an acid found in bear bile. TUDCA is a hydrophilic compound naturally produced in the liver by the union of taurine and ursodeoxycholic acid. Several studies have shown neuroprotective activities of TUDCA in HD thanks to its ability to inhibit apoptosis, attenuate oxidative stress, and reduce ER stress [282]. In a 3-NP HD rat model, TUDCA administration effectively counteracted apoptosis and reduced lesion volumes, but also protected against cognitive impairment and motor deficit [226]. Moreover, in a R6/2 HD mouse model, systemic administration of TUDCA significantly counteracted striatal neuropathology, reduced apoptosis, and improved locomotor and sensorimotor deficits [227].

### 4.14. Tacrolimus 

Tacrolimus, also called FK-506, is an immunosuppressant with anti-inflammatory and antioxidant properties commonly used to reduce allograft rejection but also in topical preparations. Tacrolimus has been suggested to exert neuroprotective effects. In two in vitro models of HD, for primary rat striatal neurons and immortalized striatal STHdh cells obtained from HD knock-in mice expressing normal or full-length mHtt, treatment with tacrolimus counteracted the negative effects of 3-NP exposure, inducing neuroprotection and reducing apoptosis [229]. In a 3-NP mouse model of HD, treatment with tacrolimus improved behavioral anomalies and restored the levels of oxidative stress markers but also levels of antioxidant enzymes such as SOD and CAT [19]. However, more studies are needed to better understand tacrolimus’ role in HD.

### 4.15. Synthetic Triterpenoids 

Synthetic triterpenoids such as 2-cyano-N-methyl-3,12-dioxooleana-1,9(11)-dien-28-oic acid (CDDO) are potent inducers of the Nrf2/ARE pathway, protecting cells from oxidative damage and suppressing the cyclooxygenase 2 and the inducible nitric oxide synthase (iNOS) [283,284,285]. Among synthetic triterpenoids, CDDO-Methyl Amide (CDDO-MA) showed to be two hundred thousand times stronger than the natural triterpenoid oleanolic acid into antioxidant functions [283,286]. After CDDO-MA oral administration, high levels of this triterpenoid are available in mouse brains compared to CDDO, which poorly penetrates the BBB. In 3-NP treated rats, treatment with CDDO-MA reduced lipid peroxidation and redox imbalance, but also reduced the loss of striatal neurons. Other synthetic triterpenoids able to upregulate the Nrf2/ARE pathway are CDDO-Ethyl Amide (CDDO-EA) and CDDO-TriFluoroEthyl Amide (CDDO-TFEA). In a N171-82Q transgenic HD mouse, the treatment with these two compounds ameliorated both the oxidative stress condition and motor impairment, promoting survival. Moreover, CDDO-TFEA and CDDO-EA counteracted the striatal atrophy in the brain and the vacuolation in the brown adipose tissue [230]. Taken together, these results suggest the promising role of synthetic triterpenoids in HD management as potent antioxidant compounds.

### 4.16. XJB-5-131 

XJB-5-131 is a synthetic compound with antioxidant properties and electron scavenging ability. Recent literature also showed a relevant role of XJB-5-131 in reducing somatic expansion and toxicity at the same time through the same redox system [287]. In a HD mouse model, XJB-5-131 administration counteracted the HD phenotype and ameliorated mitochondrial activity [231]. In HdhQ150 mice treated at different HD progression stages, XJB-5-131 administration led to reduced neuronal atrophy and improved motor function, but also a reduced level of 8-OHdG in the mitochondria and nucleus of striatal cells derived from the mice [232]. Moreover, in a R6/2 mouse model of HD, chronic treatment with XJB-5-131 ameliorated behavioral and physiological deficits in an age- and sex-dependent manner [233].

### 4.17. Probucol 

Probucol is a lipid-lowering agent with antioxidant and anti-inflammatory properties. In a HD rat model induced by 3-NP pre-treated with probucol, improved motor function and oxidative stress conditions with stimulation of GPx activity have been observed [234]. Moreover, in a YAC128 transgenic HD mouse model during the early- to mild-symptomatic stages of disease progression, chronic administration of probucol reduced the occurrence of depressive-like behaviors, highlighting a potential role for probucol in the early stages of HD [235].

### 4.18. BN82451 

BN82451 is an orally active compound that can easily reach the central nervous system, where it exerts potent neuronal protection and anti-inflammatory activity through antioxidant mechanism, sodium channel blockade, and inhibition of cyclooxygenase [288]. In a R6/2 HD mouse model, treatment with BN82451 promoted survival, improved motor function, reduced morphology loss, and reduced levels of aggregates positive to ubiquitin [236].

### 4.19. Kynurenine-3-Monooxygenase Inhibitors 

From tryptophan, the enzyme indoleamine dioxygenase produces kynurenine, which is then metabolized into different compounds, among all kynurenic acid. Kynurenic acid is a neuroprotective metabolite, the level of which has been found reduced in post-mortem brain samples of HD patients [289]. Reduced levels of kynurenic acid relate to excitotoxicity and oxidative stress induction. In this view, kynurenic acid administration in a HD *Drosophila* model counteracted neurodegeneration [290]. Hence, the administration of a kynurenine-3-monooxygenase inhibitor called JM6 has been tested, resulting in increased kynurenic acid levels and reduced extracellular glutamate in the brain. In particular, in the R6/2 HD model, chronic administration of JM6 prolonged lifespan and prevented synaptic loss [237].

## 5. The Oxidant-Antioxidant Balance: A Discussion

HD is characterized by an expansion mutation with the production of an aberrant protein that cannot be folded properly. The unfolded mHtt is responsible for numerous features of the disease, mainly impairing all those systems in which the normal Htt is involved. The presence of the aberrant protein, but also of its toxic fragments, and its interactions with each cellular component lead to overall dysfunction of the cell machinery, leading to oxidative stress. However, oxidative stress needs to be in balance with the antioxidant system to exert beneficial physiological effects. If a stress occurs and the levels of oxidants grow higher than the levels of antioxidants, a deleterious oxidative stress occurs, causing diffused damage and impaired redox signaling [291]. 

In an HD context, deleterious oxidative stress plays the major role. In fact, whether it is ER stress, UPR, mitochondrial dysfunction, or other cellular impaired systems, oxidative stress is always the common thread. The stimulation of oxidative stress take place in an amplifying circle of events in which the whole cell is entangled, until cell death occurs. In this view, to restore the oxidant–antioxidant balance becomes of critical relevance for therapeutic strategies in HD. Both enzymatic (SOD, GPx, GR, and CAT) and non-enzymatic (glutathione, CoQ10, bilirubin, L-arginine, uric acid, melatonin, transferrin, vitamins C and E, manganese, zinc, selenium, omega-3 and omega-6 fatty acids, carotenoids, and flavonoids) antioxidants were able to counteract ROS, RLS, and RNS accumulation in several models. Moreover, the recent literature raises the question of whether antioxidants are to be used alone in therapies or in combination with other compounds. Several preclinical and clinical studies showed the potential of antioxidant-based therapeutic strategies, demonstrating the neuroprotective and anti-apoptotic effects of various natural and synthetic compounds. In these studies, antioxidants counteracted oxidative stress, mainly restoring the redox balance and preventing mitochondrial dysfunction. Therefore, evidence suggested that the detrimental effects of oxidant–antioxidant imbalance play a relevant role in HD, but also that this condition can be effectively counteracted by new therapeutic strategies and approaches (Figure 2) [196,222,229,292,293].

## 6. Conclusions

Oxidative stress plays a critical role in HD. The cell machinery tries to adapt to this harmful stress by rearranging itself, aiming to compensate for damage and to restore the balance between oxidant and antioxidant molecules. However, HD pathogenesis shows specific features relatable to oxidative stress and cell machinery disruption, in which it is not already clear whether those features are the cause or the consequence of this overall dysfunction. Through the decades, the growing knowledge in this context has only been followed by more questions. Thus, new interdisciplinary approaches may be needed in the future to overcome these concerns, focusing on antioxidant molecules, the dynamics of the cell machinery, and evaluating the possibility to use therapeutic strategies based not only on the antioxidants alone but also on other genetic and cellular tools with antioxidants as an adjuvant therapy.

Methods: Extensive bibliographic research was conducted using the PubMed National Library of Medicine (NIH), Web of Science platform, Google Scholar, and Clinical Key databases. Examples of the search terms used are ‘‘Huntington’s disease’’, ‘‘antioxidants’’, ‘‘nutraceuticals’’, ‘‘oxidative stress’’, “mitochondria”, “therapeutic approach”, ‘‘in vitro’’, ‘‘in vivo’’, ‘‘clinics’’, and “experimental studies”. For screening, a restriction was made to those papers published in the last 10 years and preferably in English. Priority was given to prospective studies and reviews with adequate methodological quality. In addition, a secondary search of the bibliography of the papers finally selected was carried out to detect possible omissions. For the analysis of all relevant publications, consensus meetings were held with all the authors.

## Figures and Tables

**Figure 1 antioxidants-12-00571-f001:**
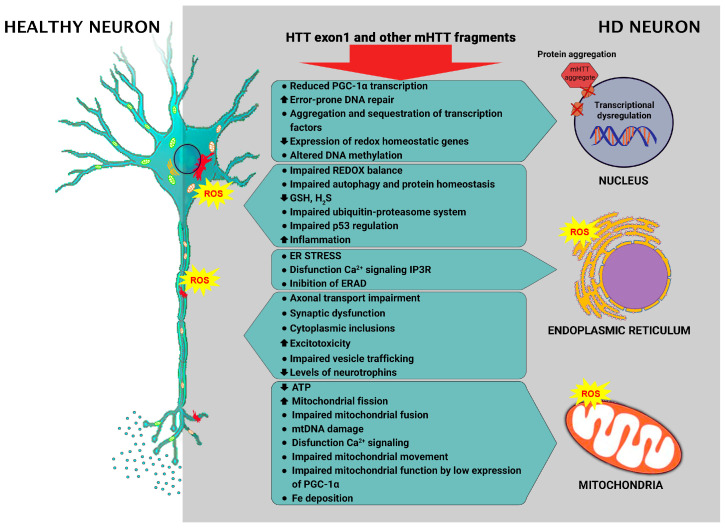
Pathogenetic mechanisms of HD responsible of neuron’s dysfunction and death.

**Figure 2 antioxidants-12-00571-f002:**
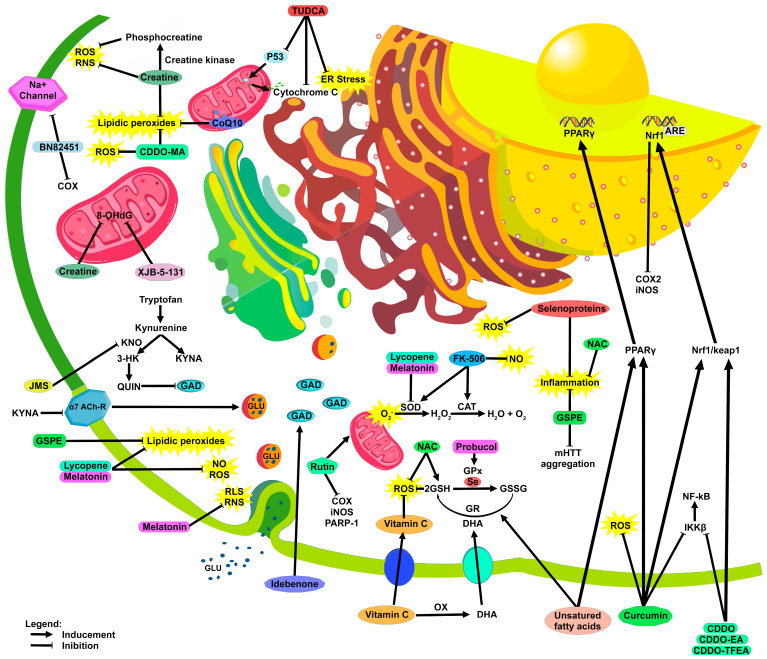
Mechanisms of action of the antioxidants discussed in the review.

**Table 1 antioxidants-12-00571-t001:** Overview of antioxidant compounds explored in this review with a potential role in HD management.

Antioxidant	Mechanism of Action	Model	Effects	References
Vitamin C	In the presence of ROS, Vitamin C quickly oxidizes to dihydroxy ascorbic acid, acting as a regulator of redox balance in neurons	R6/2 mice	Restoration of behavioral response	[188]
			Modulation of neuronal glucose uptake	[189]
		STHdhQ cells	An early impairment of ascorbic acid uptake in HD neurons.	[190]
			Modulation of neuronal glucose uptake	[189]
Selenium	Effects are mainly exerted due to the antioxidant function of selenoproteins, which help maintain the intracellular redox status and prevent cellular damage from free radicals	Quinolinic acid rat model of HD	Selenium partially protects against quinolinic acid-induced toxicity	[191]
		3-NP-induced HD-like rat model	Protective effects against HD-like signs induced by 3-NP in rats	[192]
		N171-82Q mice	Sodium selenite supplementation decreased brain weight loss and GSSG levels, but also increased motor function	[193]
Unsaturated Fatty Acids	They display antimicrobial, antioxidant, and anti-inflammatory properties	3-NP-induced HD-like rat model	EVOO act as a powerful brain antioxidant	[194]
			Decreased lipid peroxidation and GSH recovery	[195]
		Quinolinic acid rat model of HD	EVOO and fish-oil counteracted oxidative damage, increasing PPARg expression, and restoring rat behavior	[196]
		HD patients	Restoration of the motor function	[197,198]
Creatine	It is effective against superoxide, peroxynitrite, and hydroxyl radicals	R6/2 mice	A role of metabolic dysfunction in a transgenic mouse model of HD was supported, suggesting a novel therapeutic strategy to slow the pathological process	[199]
		N171-82Q mice	Creatine may exert therapeutic effects in HD	[200]
		3-NP-induced HD-like rat model	Creatine counteracted motor impairment and cognitive abnormalities	[201]
		HD patients	Creatine supplementation significantly reduced elevated serum levels of 8-OHdG back to baseline levels seen in controls	[202]
			Slowing of the ongoing cortical atrophy	[203]
			Creatine monohydrate is not beneficial for slowing functional decline	[204]
Coenzyme Q10 (CoQ10)	It acts as an electron carrier in ETC ^1^ and reduces singlet oxygen, prevents oxidation of bases in mitochondria and formation of lipid peroxyl radicals and protein oxidation	R6/2 mice	CoQ10 or remacemide significantly extended survival and delayed the development of motor deficits, weight loss, cerebral atrophy, and neuronal intra-nuclear inclusions	[205]
			The combination of CoQ10 and creatine improved motor performance and promoted survival in mice	[206]
			Creatine promoted survival of mice, improved the motor function, and counteracted the striatal atrophy	[203]
		3-NP-induced HD-like rat model	The combination treatment blocked 3-NP-induced impairment of glutathione homeostasis, reduced lipid peroxidation and DNA oxidative damage in the striatum	[206]
		HD patients	Dosages of 2400 mg/day may provide the best balance between tolerability and blood level achieved	[207]
Idebenone	It has antioxidant properties similar to CoQ10. It is a substrate of NQO 1 and 2	Kainic acid-induced rat model	Restoration of GAD immunoreactivity	[208]
		HD patients	No impact for idebenone on HD patient conditions compared to the placebo controls	[209]
Curcumin	It is involved in several functions, among which are antioxidant, neuroprotective, and anti-inflammatory activities. Regulates various pathways related to cell survivor, caspases, tumor suppression, and others	Quinolinic acid model of HD in rats	Combination of curmcumin and piperine showed strong antioxidant and protective effect against quinolinic acid-induced behavioral and neurological alteration in rats	[210]
		*Drosophila* HD model	Suppression of cell death	[211]
			Metabolic anomalies amelioration, ROS levels reduction, and motor impairment counteraction have been obtained after curcumin administration	[212]
		R6/2 mice	Overall amelioration of the HD phenotype	[213]
Grape Seed Polyphenol Extract (GSPE)	It limits lipid peroxidation and reduces inflammation	R6/2 mice	GSPE might be able to modulate the onset and/or progression of HD	[214]
		PC-12 HD	Reduced levels of carbonyl heightened by mHtt expression and inhibited formation of mHtt aggregate	
Lycopene	It exerts antioxidant effects by quenching singlet oxygen	3-NP-induced HD-like rat model	Pretreatment improved behavioral symptoms, counteracted oxidative damage, restored the activity of mitochondrial enzymes, but also reduction in lipid peroxidation, NO and SOD levels, and behavioral impairment	[215]
			Administration of lycopene in combination with other compounds lead to modulation of NO, resulting in restoration of behavioral and biochemical improvement	[216]
			Combination treatment of lycopene and quercetin with and without poloxamer 188 in HD alleviate anxiety and depression than single drug therapy	[217]
Melatonin	It shows strong antioxidant activity by scavenging ROS/ RNS and inhibiting NOS but also protects lipids in membranes, proteins in cytosol, DNA in nucleus and mitochondria from free radical damage	3-NP-induced HD-like rat model	Melatonin prevents the deleterious effects induced by 3-NP	[218]
		Quinolinic acid model of HD in rats	Melatonin helps in neurotoxicity caused by quinolinic acid in rats	[219]
		Kainic acid-induced rat model	Melatonin counteracted the oxidative damage promoting neuroprotection	[220]
N-Acetylcysteine	It reduces oxidative stress by decreasing ROS and lipid peroxidation, and by restoration of antioxidant enzymes	3-NP-induced HD-like rat model	NAC treatment may be a useful therapeutic strategy against 3-NP neurotoxicity and HD	[221]
			NAC treatment counteracted mitochondrial dysfunctions and neurobehavioral deficits	[222]
		R6/1 mice	NAC rescued glutamatergic dysfunction and depressive-like behavior in HD	[223]
Rutin	It shows antioxidant, anti-inflammatory, neuroprotective, and immunomodulatory properties.	3-NP-induced HD-like rat model	Rutin improved the behavioral alterations and restored the activities of mitochondrial complex enzymes in rat model	[224]
			Rutin may have an important role in protecting the striatum from oxidative stress caused by 3-NP	[225]
Tauroursodeoxycholic acid (TUDCA)	A hydrophilic bile acid, has anti-inflammatory and cytoprotective effects, antioxidant effects; mechanism of action not fully defined yet	3-NP-induced HD-like rat model	TUDCA administration counteracted apoptosis and reduced lesion volumes, but also protected against cognitive impairment and motor deficit	[226]
		R6/2 mice	Systemic administration of TUDCA significantly counteracted striatal neuropathology, reduced apoptosis, and improved locomotor and sensorimotor deficits	[227]
Tacrolimus (FK-506)	It has anti-inflammatory and antioxidant properties via modulation of NOS and HO activities	3-NP-induced HD-like rat model	Treatment with tacrolimus improved behavioral anomalies and restored the levels of oxidative stress markers and antioxidant enzymes such as SOD and CAT	[228]
		Primary rat striatal neurons	Treatment with tacrolimus counteracted effects of 3-NP exposure, inducing neuroprotection and reducing apoptosis	[229]
		STHdhQ cells		
Synthetic triterpenoids	Triterpenoids, particularly the analogs of CDDO, have antioxidant and anti-inflammatory properties acting via Nrf2/ARE pathway	3-NP-induced HD-like rat model	CDDO-MA can be neuroprotective in experimental model of HD	[206]
		N171-82Q mice	CDDO-EA and CDDO-TFEA upregulated Nrf2/ARE induced genes in the brain and peripheral tissues, reduced oxidative stress, improved motor impairment and increased longevity, also rescuing striatal atrophy in the brain and vacuolation in the brown adipose tissue	[230]
XJB-5-131	Synthetic compound with antioxidant properties and electron scavenging ability	HD150KI mice (test) C57BL/6 mice (controls)	XJB-5-131 can suppress weight loss, ameliorate mitochondrial activity, and improve motor performance	[231]
		HdhQ150 mice	XJB-5-131 administration led to reduced neuronal atrophy and improved motor function, but also reduced level of 8-OHdG in mitochondria and nucleus of striatal cells derived from the mice	[232]
		R6/2 mice	Chronic treatment with XJB-5-131 ameliorated behavioral e physiological deficits in an age- and sex dependent manner	[233]
Probucol	Lipid-lowering agent with antioxidant and anti-inflammatory properties	3-NP-induced HD-like rat model	Improvement in motor function and oxidative stress condition with stimulation of GPx activity	[234]
		YAC128 transgenic HD mouse model	Chronic administration of probucol reduced the occurrence of depressive-like behaviors	[235]
BN82451	It reduces excitotoxicity, oxidative stress, and inflammation and is also a mitochondrial protective agent	R6/2 mice	BN82451 promoted survival, improved motor function, reduced morphology loss, and reduced levels of aggregates positive to ubiquitin	[236]
Kynurenine-3-monooxy inhibitor: JM6	Inhibition of KMO ^2^ in blood increases kynurenic acid in brain and reduces extracellular glutamate and free radical production	R6/2 mice	Administration of JM6 resulted in increased kynurenic acid levels and reduced extracellular glutamate in the brain, prolonged lifespan and prevented synaptic loss	[237]

^1^ ETC: Electron Transport Chain; ^2^ KMO: Kynurenine Mono-Oxygenase.

## Data Availability

Not applicable.

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
