# Peer review of "Cell Rearrangement and Oxidant/Antioxidant Imbalance in Huntington’s Disease"

_antioxidants, 2023, doi:10.3390/antiox12030571_

Round 1
Reviewer 1 Report
The manuscript requires some text editing. There are several typo and grammer errors and unclear wording;
for example:
line 113: what is meant by "axonal tracking"
line 161/162: check sentence structure
line 471/472: that Q10 is involved in complex I and II activities BUT ALSO in ATP production
line 666: "... antioxidants resulted able to accomplish ...": check sentence structure
Typo errors:
line 40: "... expansion Repeats ...", full stop is missing
line 410: "... an in ....", replace by ".... a ..."
Author Response
Reviewer 1
The manuscript requires some text editing. There are several typo and grammer errors and unclear wording;
Response: We would like to thank Reviewer 1 for the time spent in reading our manuscript and for the comments provided that helped in improving our article. We appreciate the reviewer’s comments and we now thoroughly checked the manuscript as suggested. Please see lines 40, 112-113, 160-164, 440, 502, 696.
Reviewer 2 Report
The authors present a narrative review on HD, cellular processes and antioxidants. Overall the review is comprehensive but very long. A few points to consider:
-Was there a method to determine completeness of antioxidant coverage in your review (just expertise or did you do an exhaustive search)? I see the short methods section mentioning searching databases but there is no mention of inclusion/exclusion criteria, or a prisma flow diagram. The review seems to be waivering between a formal systematic review and narrative review
-Table 1 is informative but unwieldy in the document. Is there a way to break it down by category or sections to make it more readable?
-Figure 1 is great but could there by another figure highlighting the actions/MOA of the antioxidants covered? Something that drives home a summary of all antioxidants reviewed?
Author Response
Reviewer 2
The authors present a narrative review on HD, cellular processes and antioxidants. Overall the review is comprehensive but very long. A few points to consider:
Response: We would like to thank Reviewer 2 for the time spent in reading our manuscript and for the comments provided that helped in improving our article. We tried to address all the points raised.
Comments
- Was there a method to determine completeness of antioxidant coverage in your review (just expertise or did you do an exhaustive search)? I see the short methods section mentioning searching databases but there is no mention of inclusion/exclusion criteria, or a prisma flow diagram. The review seems to be waivering between a formal systematic review and narrative review
Response: We appreciate the Reviewer’s comment. We performed an exhaustive research of antioxidant molecules in HD literature, focusing on tested molecules with promising or well-defined neuroprotective effects in HD subjects. We produced a narrative review with an overview of english antioxidants literature in HD context in order to strongly highlight the therapeutic potential of antioxidant molecules in HD systems characterized by marked oxidative stress.
- Table 1 is informative but unwieldy in the document. Is there a way to break it down by category or sections to make it more readable?
Response: We thank the Reviewer for the comment and we totally agree with the Reviewer. However, we believe that an all-in-one table would produce a more immediate information than splitting the table in more sections/categories, obtaining an overall summary of the antioxidant’s usage in pre-clinical and clinical context. Moreover, in our opinion due to the type of information selected for the table it would be easier to associate them in this way.
- Figure 1 is great but could there by another figure highlighting the actions/MOA of the antioxidants covered? Something that drives home a summary of all antioxidants reviewed?.
Response: Thank you for the comment. We added a second figure that offer an overview of all the compound dissected with the respective effect/mechanism of action as suggested.
Reviewer 3 Report
The manuscript " Cell rearrangement and oxidants/antioxidants imbalance in Huntington’s Disease” by D’Egidio et al., presents a good enough English. It has a good approach to the field, but the manuscript needs major and minor revisions.
Major revision:
1.- In this illness increased mHTT expression activates immune response-mediated neurodegeneration and HTT is expressed in neurons, in advanced stages of HD the protein has been detected in B and T cells, monocytes and macrophages. Author needs to explain this fact because inflammation is involved in this disease.
2.- Exploring DNA methylation-based biomarkers in tissue age, a significant association of HD and epigenetic age acceleration and significant disruption in DNA methylation levels in brain tissue has been detected. Authors need to discuss about that.
3.- The polyglutamine expansion in the HTT protein is an indicator of early-stage HD and changes in DNA methylation have been found associated with this expansion increasing mHTT expression. Resulting of that, neurogenesis, cognition and motor function in transgenic mouse and human have been detected. Author needs to be deep of that question.
4.- In HD, dysregulation of two major transcription factor, p53 and PGC-1α, are related to induction and exacerbation of mitochondrial dysfunction, apoptosis, and neurodegeneration. This referee knows that PGC-1α is related with inflammation so indicate that in the manuscript.
Minor revision
There are some extra spaces between words that need to be corrected.
Author Response
Reviewer 3
The manuscript " Cell rearrangement and oxidants/antioxidants imbalance in Huntington’s Disease” by D’Egidio et al., presents a good enough English. It has a good approach to the field, but the manuscript needs major and minor revisions.
Response: We would like to thank Reviewer 3 for the positive comments and the time spent in reading our manuscript. We would like to thank Reviewer 3 for the time spent in reading our manuscript and for the comments provided that helped in improving our article. We tried to address all the points raised.
Major revision
- In this illness increased mHTT expression activates immune response-mediated neurodegeneration and HTT is expressed in neurons, in advanced stages of HD the protein has been detected in B and T cells, monocytes and macrophages. Author needs to explain this fact because inflammation is involved in this disease.
- In HD, dysregulation of two major transcription factor, p53 and PGC-1α, are related to induction and exacerbation of mitochondrial dysfunction, apoptosis, and neurodegeneration. This referee knows that PGC-1α is related with inflammation so indicate that in the manuscript.
Response: We appreciate the Reviewer’s comments 1 and 4, and we agree. We added information as suggested. Please, see lines 204-209 and 376-384.
- Exploring DNA methylation-based biomarkers in tissue age, a significant association of HD and epigenetic age acceleration and significant disruption in DNA methylation levels in brain tissue has been detected. Authors need to discuss about that.
- The polyglutamine expansion in the HTT protein is an indicator of early-stage HD and changes in DNA methylation have been found associated with this expansion increasing mHTT expression. Resulting of that, neurogenesis, cognition and motor function in transgenic mouse and human have been detected. Author needs to be deep of that question.
Response: We thank the reviewer 3 for the comments 2 and 3. We added the missing information as suggested. Please, see lines 169-191.
Minor revision
There are some extra spaces between words that need to be corrected.
Response: Thank you for the comment, we carefully checked the manuscript also using Ginger software.
Round 2
Reviewer 3 Report
Accept in present form